# Multi-Feature-Filtering-Based Road Curb Extraction from Unordered Point Clouds

**DOI:** 10.3390/s24206544

**Published:** 2024-10-10

**Authors:** Hong Lang, Yuan Peng, Zheng Zou, Shengxue Zhu, Yichuan Peng, Hao Du

**Affiliations:** 1The Key Laboratory for Traffic and Transportation Security of Jiangsu Province, Huaiyin Institute of Technology, Huaian 223003, China; gglehonglang@gmail.com; 2The Key Laboratory of Road and Traffic Engineering of the Ministry of Education, Tongji University, Shanghai 201804, China; 2231323@tongji.edu.cn (Y.P.); zouzhn@tongji.edu.cn (Z.Z.); yichuanpeng1982@hotmail.com (Y.P.); 3Shanghai Tongke Transportation Technology Co., Ltd., Shanghai 200092, China

**Keywords:** LiDAR, unordered point clouds, curb extraction, multi-feature filter

## Abstract

Road curb extraction is a critical component of road environment perception, being essential for calculating road geometry parameters and ensuring the safe navigation of autonomous vehicles. The existing research primarily focuses on extracting curbs from ordered point clouds, which are constrained by their structure of point cloud organization, making it difficult to apply them to unordered point cloud data and making them susceptible to interference from obstacles. To overcome these limitations, a multi-feature-filtering-based method for curb extraction from unordered point clouds is proposed. This method integrates several techniques, including the grid height difference, normal vectors, clustering, an alpha-shape algorithm based on point cloud density, and the MSAC (M-Estimate Sample Consensus) algorithm for multi-frame fitting. The multi-frame fitting approach addresses the limitations of traditional single-frame methods by fitting the curb contour every five frames, ensuring more accurate contour extraction while preserving local curb features. Based on our self-developed dataset and the Toronto dataset, these methods are integrated to create a robust filter capable of accurately identifying curbs in various complex scenarios. Optimal threshold values were determined through sensitivity analysis and applied to enhance curb extraction performance under diverse conditions. Experimental results demonstrate that the proposed method accurately and comprehensively extracts curb points in different road environments, proving its effectiveness and robustness. Specifically, the average curb segmentation precision, recall, and F1 score values across scenarios A, B (intersections), C (straight road), and scenarios D and E (curved roads and ghosting) are 0.9365, 0.782, and 0.8523, respectively.

## 1. Introduction

Environmental perception technology is a vital component of autonomous vehicle technology, significantly impacting the safety of autonomous vehicles [1]. The environmental perception system of an autonomous vehicle perceives environmental information to provide a foundation for vehicle decision making and control [2]. Road edge detection, as an important part of vehicle environmental perception, aims to utilize vehicle-mounted perception equipment to determine road boundaries, thereby distinguishing the road from the background [3]. The accurate identification of road boundaries and extraction of road surface information are foundational for the realization of vehicle driver assistance systems and are two of the key technologies for autonomous vehicle navigation [4].

With the development of computer vision technology, some researchers have employed video sensors to detect road edges from video images. Florin Oniga et al. [5] utilized edge detection based on video images and Hough transform to extract road edges. Wang et al. [6] used a naive Bayesian transformation to fuse multiple image features, calculating the probability that each point is a curb point. Additionally, some researchers have employed data mining techniques to extract road edges from driving video sets [7]. However, identifying curbs based on video images is susceptible to environmental factors such as weather and lighting conditions and performs poorly when there are cracks or shadow interferences [7,8]. Moreover, due to the large amount of computation required to process video data, it is difficult to ensure real-time detection. LiDAR has high accuracy and is less susceptible to environmental interference compared to other methods; hence, many studies consider using LiDAR for road edge extraction.

Commonly used LiDAR methods can be categorized into two-dimensional (2D) LiDAR and three-dimensional (3D) LiDAR. Himstedt et al. [9] used data collected by 2D LiDAR and employed geometrical landmark relations (GLARE) for large-scale place recognition. However, compared to 3D LiDAR, the point cloud from 2D LiDAR is relatively sparse and lacks height information, making it challenging for 2D LiDAR to meet the environmental perception requirements of autonomous vehicles [10]. Therefore, 3D LiDAR has become the primary tool for environmental perception using LiDAR.

Road information collection using LiDAR mainly relies on airborne laser scanning (ALS) [11] and mobile laser scanning (MLS) [12]. However, because airborne laser scanning is typically suitable for large-scale detection, it struggles to reflect the finer details of curbs when scanning road edges, affecting the accuracy and completeness of curb extraction. Consequently, most current research utilizes mobile laser scanning to collect curb point clouds. MLS comprises a laser scanner, Global Navigation Satellite System/Inertial Measurement Unit (GNSS/IMU) system, and digital cameras. It generates a 3D point cloud by recording the geometric shape and intensity information of a scene and captures color/texture information generated by the digital cameras [12].

Extracting curbs from point clouds collected by MLS has become a popular research topic. Current curb detection methods mainly fall into two categories: the planarization of 3D point clouds and methods based on curb features.

### 1.1. 3D Point Cloud Gridding

3D point cloud gridding involves dividing a plane into multiple grids and projecting the point cloud onto this plane. Subsequently, features such as the height, intensity, and normal vectors are extracted from each point in the 3D point cloud and associated with the grids to detect and identify curbs.

Yin et al. [13] proposed a double-grid algorithm with adaptive threshold generation. The point cloud is divided into grids based on plane coordinates and the improved Otsu method is used to adaptively generate thresholds. Curb detection is then achieved by applying threshold constraints within and between adjacent grids. Yue et al. [14] used a method of statistical point cloud height distribution within grids for obstacle classification that avoided the impact of noise points. They then used edge point search methods to detect road edges, achieving ideal results even with interference from vehicles, branches, and other objects. Yang et al. [15] automatically extracted structured and unstructured road edges based on laser scanning lines and grid networks. This method was based on surface roughness and did not consider height, slope, or density threshold methods. Jaakkola et al. [16] generated raster images using reflection intensity and height attributes from point clouds and extracted curbs based on the differences in height and intensity between curb points and road points. Hernández et al. [17] first projected 3D point clouds onto a 2D plane, then used quasi-flat region algorithms and region adjacency graphs to extract edge points from point cloud images, selecting those with significant differences from the road (approximately 14 cm). Serna et al. [18] mapped point clouds to depth images and used height and geodetic features to detect curb areas. This method was tested on MLS databases from Paris (France) and Enschede (Netherlands), achieving high detection accuracy with few false positives. However, although the point cloud gridding method is mature, efficient, and simple, the gridding process can cause some feature loss, often failing to truly reflect the original road conditions. Therefore, some scholars directly use the 3D features of the curb to extract curbs.

### 1.2. Curb Extraction Based on 3D Features

Methods based on 3D curb features directly extract curb points from the 3D point cloud using their 3D characteristics. Wang et al. [19] extracted road edges based on the vertical and linear features of curbs using local attributes such as slope differences. Yang et al. [20] divided the MLS point cloud into a set of continuous “scanning lines”, each containing a road cross-section. They detected curbs using height differences, point densities, and slopes with a moving window algorithm. Smadja et al. [21] processed MLS data using the RANSAC (Random Sample Consensus) algorithm and represented the road plane with polynomials. Yuan et al. [22] segmented point cloud images using a fuzzy clustering method based on maximum entropy theory, then used the least-squares method to fit the road edges. Kim et al. [23] used the Hough transform method to find the best-fit line for the road and then used the endpoints of this line as curb points. Li Guangjing et al. [24] first obtained candidate edge points based on height and smoothness features, then used the RANSAC algorithm to fit polynomials to the curb points on both sides and finally used the Kalman filter for edge point prediction and tracking. Methods based on 3D curb features can make the most of the spatial characteristics of curbs, improving the accuracy and completeness of curb extraction. However, they rely on standard curb features as screening and reference bases, affecting the accuracy and completeness of extraction in complex scenarios such as intersections.

3Dpoint clouds are categorized into ordered and unordered types. Current curb point cloud extraction methods are based on ordered point clouds. Ordered point clouds have points organized in a specific sequence, with each point’s position information associated with its index in the point cloud, having a clear topological structure. They are usually used to represent structured data obtained through scanning or modeling [3,15,25,26]. Unordered point clouds have no specific order, and each point’s position information is independent of other points, without a clear topological structure. Unordered point clouds are directly sampled from real-world objects or scenes, accurately capturing the shapes and details of real objects in their original forms, avoiding information loss due to explicit topological connections or triangulation [1,4,19]. Additionally, unordered point clouds do not require pre-defined topological structures, making them suitable for irregular shapes, complex geometries, and diverse scenes. When dealing with real-world objects, unordered point clouds can adapt more freely to various shapes and geometrical structures. However, most of the current research on curb extraction using point clouds is based on ordered point clouds, and studies on using unordered point clouds for curb extraction are still in the exploratory stage, leading to poor generalization performance in complex scenarios.

This paper studies curb extraction in various complex scenarios based on 3D unordered point clouds. Initially, curb points are rapidly and coarsely extracted using grid height differences and normal vector features. Building on this, innovative methods such as clustering, a density-based alpha-shape algorithm, and the MSAC (M-Estimate Sample Consensus) algorithm based on multi-frame fitting are employed for the fine extraction of curb points. Sensitivity analysis is conducted to determine the optimal threshold values for various features in the multi-feature filter. Utilizing this multi-feature filter, accurate, complete, and robust curb extraction is achieved in complex scenarios, such as in intersections. The main contributions are as follows:This paper proposes a multi-feature filtering framework for curb extraction from unordered point clouds. This framework integrates several techniques, including the grid height difference, normal vectors, clustering, a density-based alpha-shape algorithm, and a multi-frame fitting approach using the MSAC algorithm to accurately identify curbs in complex road scenarios.This paper introduces a multi-frame MSAC fitting approach that improves on traditional single-frame methods by fitting the curb contour every five frames. This approach captures the full curb contour more accurately while preserving local features. Based on our self-developed dataset and the Toronto dataset, our method outperforms existing approaches, such as Mi’s method, in both precision and recall, demonstrating its robustness across various complex road scenarios.This paper performs parameter sensitivity analysis across different road scenarios to determine the range of parameter values that do not significantly influence the curb segmentation results, further validating the robustness of the method.

The main content is as follows. The Section 2 elaborates on the curb extraction model. The Section 3 introduces the data acquisition equipment and tests the curb extraction model in various complex road scenarios, determining the optimal threshold values through sensitivity analysis. The Section 4 summarizes the study and discusses future prospects.

## 2. Methodology

The algorithm framework is shown in Figure 1. The extraction of road curbs from unordered point clouds consists of five steps. First, the preprocessed MLS data are filtered based on grid height differences. Second, the angle between the normal vector of each point and the Z-axis is calculated to select points that meet the angle criteria. Third, DBSCAN is used for clustering, retaining clusters whose intra-cluster distances meet the threshold requirements. Fourth, the alpha-shape algorithm based on point cloud density is employed to retain all points that the rolling circle passes through. Finally, the multi-frame MSAC algorithm is used to calculate the distance of each point from the fitting plane, retaining all points whose distances are below the threshold; these points are the final curb points.

### 2.1. Grid Height Difference

The data in unstructured point clouds include the X, Y, and Z coordinates of each collection point. First, the point cloud is leveled and rotated, setting the ground plane as the XOY plane of the point cloud, with the Z coordinate representing the height of each point relative to the ground plane. The coordinates of each point in the 3D point cloud are denoted as $p_{k}$(*x^k^*, *y^k^*, *z^k^*), where *k* = 1, 2, 3, ….

The XOY plane of the 3D coordinate system is divided into a grid network (numGridsx × numGridsy) where the smallest unit of the grid network is a square grid with side length *l* × *l.* The grid network’s dimensions, its rows and columns, represent the numbers of square grids in the x and y directions, respectively. The rows and columns of the grid network are calculated as shown in the following equations:(1)$$numGridsx=\left\lceil \left.\frac{\max\left(x^{k}\right)-\min\left(x^{k}\right)}{l}\right\rceil \right.,\;\;k=1,\;2,\;3,\ldots ,n$$
(2)$$numGridsy=\left\lceil \left.\frac{\max\left(y^{k}\right)-\min\left(y^{k}\right)}{l}\right\rceil ,\;\;k=1,\;2,\;3,\ldots ,n\right.$$

In Equations (1) and (2), numGridsx and numGridsy represent the numbers of rows and columns of the grid, respectively. *n* is the number of points in the point cloud, *l* is the grid length, $x^{k}\;and\;y^{k}$ are the coordinates of each point, and ⌈ ⌉ denotes the ceiling function, which rounds up to the nearest integer.

Next, each point in the 3D coordinate system is projected vertically onto the corresponding grid on the XOY plane. Each point (*i*, *j*) is assigned to a specific grid area based on its x and y coordinates.
(3)$$i=\frac{\left\lfloor x^{k}-\min\left(x^{k}\right)\right\rfloor }{l}+1,\;\;k=1,\;2,\;3,\ldots ,n$$
(4)$$j=\frac{\left\lfloor y^{k}-\min\left(y^{k}\right)\right\rfloor }{l}+1,\;\;k=1,\;2,\;3,\ldots ,n$$

In Equations (3) and (4), *i* is the x-direction coordinate of the grid that point $p_{k}$(*x^k^*, *y^k^*, *z^k^*) belongs to in the grid network and *j* is the y-direction coordinate. The symbol ⌊ ⌋ represents the floor function. Using the above Formulas (1)–(4), each point *p*($x_{\left(i,j\right)}^{k},\;y_{\left(i,j\right)}^{k}$, $z_{\left(i,j\right)}^{k}$) is assigned to a specific square grid, with the subscript (*i*, *j*) representing the grid’s position in the grid network.

To avoid the influence of outliers on the calculations, points whose Z coordinates fall outside the mean ± 3 standard deviations in each square grid are removed. Then, the Z coordinates of the points within each square grid are sorted, denoted as follows:(5)$$z_{\left(i,j\right)}^{1}>z_{\left(i,j\right)}^{2}>z_{\left(i,j\right)}^{3}>\ldots >z_{\left(i,j\right)}^{m-1}>z_{\left(i,j\right)}^{m}$$

Here, *m* is the total number of points in the square grid. The height difference for each grid, denoted as ${∆H}_{\left(i,j\right)}$, is obtained using the relevant formula.
(6)$${∆H}_{\left(i,j\right)}=\frac{1}{s}[\sum_{k=1}^{s}\;z_{\left(i,j\right)}^{k}-\sum_{k=m-s}^{m}\;z_{\left(i,j\right)}^{k}]$$

In Equation (6), ∆H represents the height difference of each grid, $z_{\left(i,j\right)}^{k}$ denotes the Z-coordinate of the points within the square grid, and *s* is a set constant.

In urban roads, curbs usually have a certain height, which varies by country and region but generally ranges from 10 to 30 cm. In the curb area, the Z-axis height changes suddenly. Therefore, if a grid contains curb points, its height difference ${∆H}_{\left(i,j\right)}$ will be greater than that of a grid containing only ground points. Based on this principle, grids with a height difference greater than 10 cm are selected as containing curb points. Therefore, the condition for judging whether a point in the point cloud is a curb point using the grid height difference is as shown in Equation (7).
(7)$$H_{thr1}<\frac{1}{s}\left[\sum_{k=1}^{s}\;z_{\left(i,j\right)}^{k}-\sum_{k=m-s}^{m}\;z_{\left(i,j\right)}^{k}\right]<H_{thr2}$$

In Equation (7), $H_{thr1}$ and $H_{thr2}$ are the height difference thresholds, set at 5 cm and 30 cm, respectively. Compared to directly using the height of each point for filtering, using the grid height difference for coarse extraction effectively avoids the interference of local road protrusions and depressions on the results. After removing outliers, the grid height difference, calculated as the difference between the average elevation of the highest *s* points and the lowest *s* points, better represents the elevation differences within the grid.

After processing the grid height differences, some ground points are preliminarily filtered out from the 3D laser point cloud image while completely retaining the road curbs. However, many points that do not belong to curbs still pass the filter, necessitating further point cloud processing steps.

### 2.2. Normal Vector

Unstructured unordered point clouds only contain the 3D coordinates X, Y, and Z for each point. The normal vector for each point is obtained using the k-nearest-neighbor (k-NN) method based on the 3D coordinates of the point cloud [26].

Since the surface of the curb is approximately perpendicular to the ground, the angle between the normal vector of curb points and the Z-axis is around 90 degrees. On the other hand, ground points, which coincide with the XOY plane, have normal vectors with angles close to 0 degrees relative to the Z-axis. Therefore, curb points can be further extracted based on the angle between the normal vector and the Z-axis. Thus, the condition for determining whether a point is a curb point based on the normal vector is as shown in Equations (8) and (9).
(8)$$\theta_{<{\vec{n}}_{p_{k}},\vec{z}>}={cos}^{-1}\frac{{\vec{n}}_{p_{k}}\vec{z}}{\left\|{\vec{n}}_{p_{k}}\right\|\left\|\vec{z}\right\|}$$
(9)$$\left|\theta_{<{\vec{n}}_{p_{k}},\vec{z}>}-90^{{}^{\circ}}\right|<\theta_{thr}$$

In Equations (8) and (9), ${\vec{n}}_{p_{k}}$ represents the surface normal vector of a point in the point cloud; $\vec{z}$ represents the direction vector of the Z-axis, taken as (0, 0, 1); and $\theta_{<{\vec{n}}_{p_{k}},\vec{z}>}$ is the angle between the normal vector ${\vec{n}}_{p_{k}}$ and $\vec{z}$. $\left\|{\vec{n}}_{p_{k}}\right\|\left\|\vec{z}\right\|$ represents the magnitude of vectors ${\vec{n}}_{p_{k}}\;$ and $\vec{z}$ and $\left|\theta_{<{\vec{n}}_{p_{k}},\vec{z}>}\right|$ is the absolute value of the angle. The angle threshold $\theta_{thr}$ for determining whether a point is a curb point was set at 35 degrees in this paper.

After filtering based on normal vectors, most ground points are removed and curb points are retained. However, some non-curb interference points are also extracted. At this stage, further refinement using clustering and RANSAC filtering is needed to accurately and completely extract the road curbs.

### 2.3. Clustering

Since road edges and obstacles (such as trees and vehicles) have significantly different attributes, distinguishing between different clusters after clustering can further accurately filter out obstacle points. Road edges are typically elongated shapes with larger intra-cluster distances when measured by Euclidean distance whereas obstacles have more irregular shapes and relatively smaller intra-cluster distances compared to road edges. Thus, this feature can be used to extract road edges and precisely filter out obstacle points.

Since the shapes of road edges and obstacles may be non-convex, a density-based clustering method can handle non-convex shapes effectively. Additionally, density-based clustering algorithms are noise-tolerant and do not require pre-setting the number of clusters. Therefore, the DBSCAN algorithm is used for clustering the point cloud [27].

(1)Parameter Setting

The DBSCAN algorithm has two key parameters: the neighborhood radius ε and the minimum number of points *minPts*. The radius ε is used to define the ε-neighborhood of a point, where points within the ε distance of the point are considered its neighbors. The minimum number of points *minPts* is used to determine a core point. If the number of points in a point’s ε-neighborhood (including the point itself) is greater than or equal to *minPts*, the point is considered a core point.

(2)Core Point Marking

Each point in the dataset is traversed. If the number of points in its ε-neighborhood is greater than or equal to *minPts*, the point is marked as a core point and its neighbors are added to the current cluster. If the point is not a core point but is within the ε-neighborhood of a core point, it is marked as a border point.

(3)Cluster Expansion:

Starting from a core point, the point is added to the current cluster. Each point in the ε-neighborhood of the core point is traversed. If a point is a core point and has not been visited, it is added to the current cluster and its ε-neighborhood points are further expanded. If a point is a border point and has not been visited, it is added to the current cluster but its neighborhood is not expanded. This process is repeated until the ε-neighborhoods of all core points in the current cluster have been traversed.

(4)Noise Point Handling

All points that have not been visited are considered noise points and are marked as part of a separate cluster.

After obtaining the road point cloud clusters ${cl}_{1},\;{cl}_{2}$, …, ${cl}_{N}$ (with N being the total number of clusters), the cluster center coordinates $o_{cl}$ for each cluster *cl* are calculated using Equation (10). The average distance $d_{po}$ of each point from the center coordinates $o_{cl}$ is calculated using Equation (11). If the distance $d_{po}$ satisfies Equation (12), the cluster is retained.
(10)$$o_{cl}=(\frac{1}{r_{cl}}\sum x^{k},\;\frac{1}{r_{cl}}\sum y^{k},\;\frac{1}{r_{cl}}\sum z^{k})$$
(11)$$d_{po}=\frac{1}{r_{cl}}*\sum_{k=1}^{r_{cl}}\sqrt{{\left(x^{k}-\frac{1}{r_{cl}}\sum x^{k}\right)}^{2}+{\left(y^{k}-\frac{1}{r_{cl}}\sum y^{k}\right)}^{2}+{\left(z^{k}-\frac{1}{r_{cl}}\sum z^{k}\right)}^{2}}$$
(12)$$d_{po}>D_{thr}$$

In Equation (11), $r_{cl}$ represents the total number of points in a cluster and $x^{k},\;y^{k},\;and\;z^{k}$ are the x, y, and z coordinates of the points. The distance threshold $D_{thr}$ is set to 2 m. By using the maximum intra-cluster distance, road edge point clouds can be filtered out. Additionally, since the left and right curbs are far apart, they will be divided into two different clusters. Thus, clustering can also separate the left and right curbs, providing a basis for subsequent MSAC filtering.

### 2.4. Alpha-Shape Algorithm Based on Point Cloud Density

The alpha-shape algorithm is a computational geometry method used to extract geometric shapes from point cloud data [28]. It is primarily utilized to identify and extract bounded regions within point clouds, which may represent the surfaces or boundaries of actual objects. The alpha-shape algorithm is applied in various fields. Its basic concept involves determining the boundary of a point cloud by rolling an imaginary fixed-radius circle over it; the points touched by this rolling circle form the boundary of the point cloud.

However, the traditional alpha-shape algorithm cannot adjust the radius of the rolling circle. This limitation can lead to the omission of some boundary points when the point cloud is densely distributed as the fixed-radius circle might be too large. Conversely, when the point cloud is sparsely distributed, a small radius might miss points that are farther apart. To achieve the complete and robust extraction of road edges, we have designed an alpha-shape algorithm based on point cloud density.

(1)Determining the Neighboring Point Set

For a point $p_{k0}$ that the rolling circle touches, we calculate the Euclidean distances to all other points from $p_{k0}$. We select the ten points with the smallest Euclidean distances as the neighboring point set $p_{set}$ (excluding $p_{k0}$). W denote these points as $p_{k1}$,$\;p_{k2}$,…,$\;p_{k10}$.

(2)Determining the Radius $\alpha_{p_{k0}}$ of the Rolling Circle

As shown in the following equation, we compute the Euclidean distances from each point in the neighboring point set, from $p_{set}$ to $p_{k0}$, and calculate the average of these distances. This average is used as the radius $\alpha_{p_{k0}}$ for the alpha-shape algorithm’s rolling circle.
(13)$$\alpha_{p_{k}}=\frac{1}{10}\sum_{i=1}^{10}\sqrt{{\left(p_{ki}^{x}-p_{k0}^{x}\right)}^{2}+{\left(p_{ki}^{y}-p_{k0}^{y}\right)}^{2}+{\left(p_{ki}^{z}-p_{k0}^{z}\right)}^{2}}$$

In Equation (13), $p_{ki}^{x}$, $p_{ki}^{y}$, and $p_{ki}^{z}$ represent the x, y, and z coordinates of points in $p_{set}$ and $p_{k0}^{x}$, $p_{k0}^{y}$, and $p_{k0}^{z}$ represent the x, y, and z coordinates of $p_{k0}$.

By implementing this adaptive approach to determine the rolling circle’s radius based on point cloud density, we can more accurately and comprehensively extract road edges from point cloud data.

### 2.5. MSAC Filtering Algorithm Based on Multi-Frame Fitting

After clustering, some obstacle points near the road edge might be misclassified as edge points. To accurately and completely extract the road boundary, we employ the MSAC algorithm for the final road edge extraction. MSAC is an improved version of the Random Sample Consensus (RANSAC) algorithm. RANSAC is used to estimate models with certain geometric features from point cloud data by fitting a plane model and filtering out outliers. The steps of the RANSAC algorithm are as follows [29]:(1)Random Sample Selection: The model randomly selects a small subset of sample points from the original point cloud.(2)Model Estimation: It estimates the model parameters using the selected sample points. For example, it uses the least squares method to fit a road edge plane model.(3)Inlier Selection: It calculates the distance $x_{p}$ of all points from the estimated model. It marks points with distances lower than a given threshold $x_{0}$ as inliers. These inliers are considered to fit the model while points with larger distances are treated as outliers.(4)Inlier Count Judgement: It counts the inliers selected in step 3 and determines whether their number meets a pre-set threshold *N*. If the number of inliers meets or exceeds the threshold, the estimated model is considered valid and the process moves to the next step. Otherwise, the process returns to step 1 and reselects samples.(5)Output: The algorithm repeats the above steps until a plane model meeting the distance threshold $x_{0}$ and inlier number threshold *N* is estimated. All inliers that fit the estimated model are output as the final road edge points.

However, the selection of thresholds in the RANSAC algorithm significantly impacts the fitting results. When the threshold is too large, too many points are considered inliers (road edge points), and when it is too small, some road edge points are excluded, compromising the extraction completeness. Therefore, compared to the RANSAC algorithm, the MSAC algorithm optimizes the model’s judgment criteria: if the distance $x_{p}$ from a point to the plane is lower than the threshold $x_{0}$, it is considered an inlier with a weight of $x_{p}$; otherwise, its weight is $x_{0}$. The plane with the smallest overall weight is chosen as the fitting plane and all points within the threshold distance to the fitting plane are retained. This method more precisely measures the relationship between the point-to-plane distance and the threshold, reducing the impact of threshold selection on the model compared to the RANSAC algorithm.

Currently, the MSAC filtering algorithm typically applies to point cloud data from a single frame. However, a single frame’s data, due to its short range, may not accurately reflect the actual variations of the road edge, leading to discrepancies between the fitted road edge points (inliers) and the true road edge points. To address this issue, we propose extracting all points from every *f*_0_ (where *f*_0_ = 5) frames of point cloud data and applying the MSAC algorithm for fitting. This method involves multi-frame fitting with the MSAC algorithm. The detailed procedure is shown in Algorithm 1.
**Algorithm 1:** MSAC Filtering Algorithm Based on Multi-Frame Fitting  Input: *F* frames after previous steps, where each frame contains points *P_f_*.  Output: Road curb from *F* frames *Curb_F_*.
f = 1$x_{0}$ = 0.1Initialize the collection of points from 5 frames P_5_ = {}Initialize the collection of curb points from 5 frames Curb_5_ = {}Initialize the collection of curb points from F frames Curb_F_ = {}for f = 1 to F do  P_5_ = P_5_ + P_f_  Use P_5_ to compute the MSAC fitting plane and record the inlier points set as Pinner   Curb_5_ = Pinner  if f mod 5 equals 0, do    Curb_F_ = Curb_F_ + Curb5    Curb_5_ = {}    P_5_ = {}return Curb_F_

## 3. Experimental Results and Analysis

### 3.1. Experimental Dataset

Figure 2 illustrates the self-constructed MLS system for field data acquisition [1,4]. The Livox Horizon 3D LiDAR (hereafter referred to as Horizon) is used to collect road point cloud data. The LiDAR has a wavelength of 905 nm, a horizontal field of view (FOV) of 81.7°, and a vertical FOV of 25.1°. Horizon utilizes Livox’s independently developed high-speed non-repetitive scanning technology and a custom-designed multi-line packaged laser, enabling the rapid capture of scene details. Horizon can be configured in single/double echo modes, with a point cloud data rate reaching up to 480,000 points per second in double echo mode. Horizon’s adaptability to the environment is strong; even under intense sunlight interference of 100 klx, the noise rate remains below 0.01%. The INS sensor provides essential GPS positioning and attitude data. The MLS system is mounted on a moving vehicle to scan road environment information and outputs the obtained road point cloud.

As shown in Figure 3, multiple road point clouds were collected in Shanghai, China, including various road scenarios such as straight roads, curved roads, and intersections. Figure 3a depicts the point cloud data collected from a 4.6 km intersection road segment in Shanghai with four lanes in both directions, where A and B are “T” shaped intersections. Figure 3b shows the point cloud data collected from a 1.5 km straight road segment in Shanghai with four lanes in both directions, where C represents the straight road segment. Figure 3c illustrates the point cloud data collected from an 0.8 km curved road segment with vehicle “ghosting” interference, where E represents the segment with ghosting interference. As indicated by D, this road segment has a curb on only one side and the curb is located on the curved segment.

### 3.2. Results and Analysis

To validate the effectiveness and robustness of the extraction algorithm, the proposed unordered point cloud extraction algorithm based on multi-feature filtering was applied to the collected road segments for curb extraction. Additionally, the true curbs were manually labeled in each frame. Based on the threshold sensitivity analysis in Section 3.3, the algorithm’s threshold values are shown in Table 1.

The algorithm for extracting road curbs consists of five steps: finding grid height difference, normal vector extraction, clustering, using variable-radius alpha-shape algorithm, and using multi-frame fitting MSAC algorithm. Experiments were conducted on the road point clouds collected as shown in Figure 3. The extraction results for each step are illustrated in Figure 4, Figure 5 and Figure 6. Steps (a) to (d) represent the results after processing through the grid height difference, normal vector extraction, clustering and variable-radius alpha shape algorithm, and multi-frame fitting MSAC algorithm, respectively.

As shown in Figure 4a, Figure 5a and Figure 6a, due to the presence of vehicles and roadside greenery in the road point clouds, there were significant height differences between these interference objects and the ground, resulting in numerous noise points around the curbs. Figure 4b, Figure 5b and Figure 6b demonstrate that after filtering with the normal vector, most noise points were removed because they did not meet the threshold conditions of the normal vector. Figure 4c, Figure 5c and Figure 6c show that distant obstacles such as vehicles and pedestrians were filtered out due to their distances from the curbs and small intra-cluster distances after clustering. Finally, the multi-frame fitting MSAC algorithm produced the actual road curbs, as shown in Figure 4d, Figure 5d and Figure 6d. In conclusion, the constructed multi-feature filter accurately and robustly extracted the true boundaries of complex road environments.

Traditional curb extraction methods typically use MSAC fitting for single-frame point clouds [1]. However, since a single frame contains only a short length of the curb, it often fails to accurately reflect the actual contour variations of the curb, resulting in fitted contours that may significantly deviate from those of the true curb. To address this issue, we innovatively adopted a multi-frame fitting approach, fitting the curb contour every five frames. This method captures the true curb contour while retaining local features of the curb as much as possible.

As shown in Figure 7, Figure 7a,c,e depict the results of road extraction using single-frame fitting. Since the fitted curve from a single frame does not represent the true curb curve, the MSAC algorithm includes many interference points as inliers, leading to a noisy extraction result. In contrast, Figure 7b,d,f show the results of road edge extraction using multi-frame fitting. This approach eliminates the interference of road obstacles and vehicles, accurately and completely extracting the road curb.

To quantitatively evaluate the algorithm, the true curbs were manually labeled for four scenarios (A, B, C, D, and E). Three evaluation metrics were introduced [30,31,32]: precision, recall, and the F1 score. Precision indicates the proportion of true curb points among the detected curb points, recall indicates the proportion of correctly detected curb points among the manually labeled curb points, and the F1 score represents the harmonic mean of the precision and recall [32].
(14)$$precision=\frac{TP}{TP+FP}$$
(15)$$recall=\frac{TP}{TP+FN}$$


(16)
$$F1-score=\frac{2*precision*recall}{precision+recall}$$


In Equations (14)–(16), *TP* denotes the number of true positives, *FP* denotes the number of false positives, and *FN* denotes the number of false negatives. The results of precision, recall, and the F1 score for scenarios A, B, C, D, and E are shown in Table 2. In the scenarios A, B (intersections), and C (straight road) and scenarios D and E (curved roads and ghosting), the constructed multi-feature filter performed well, with F1 scores consistently above 0.8, demonstrating the strong robustness of the algorithm. As shown in Table 2, the performance of the method proposed in this paper is compared with that of Reference 31 in the ABCDE scenario. When the scenario is relatively simple, such as in the straight-line scenario C, the F1 score of the method proposed in this paper is slightly lower than that of Mi’s method [31], with a small difference in results. However, when the scenario is more complex, such as in A and B (intersections) and in D and E (curved roads and ghosting), the F1 score of the method proposed in this paper is higher and the extraction effect of the road edges is better. This demonstrates that the method proposed in this paper has strong scene robustness.

To further demonstrate the effectiveness of the method proposed in this paper, additional experiments were conducted on Toronto dataset. As shown in Figure 8, a section of the road with a length of 3 km was selected and experiments were performed using both the method proposed in this paper and Mi’s method. As shown in the red box, the proposed method effectively reduces noise and maintains robustness, ensuring more complete curb point detection compared to Mi’s method. The results are shown in Table 3. Our proposed method outperformed Mi’s results in both precision and recall rates, indicating that the method presented in this paper can accurately and completely achieve curb extraction.

### 3.3. Sensitivity Analysis of Parameters

#### 3.3.1. $H_{thr1}$ and $H_{thr2}$

$H_{thr1}$ is the minimum height difference for a grid to be retained and $H_{thr2}$ is the maximum height difference for a grid to be retained. If $H_{thr1}$ is set too low, too many points will meet the condition, leading to excessive processing times and insufficient noise filtering. Conversely, if $H_{thr1}$ is set too high, some curbs may be excluded. $H_{thr2}$ determines the maximum height difference for a grid to be retained, distinguishing curbs from taller obstacles such as vehicles, trees, and greenery.

To analyze the relationship between the algorithm’s extraction results and the values of $H_{thr1}$ and $H_{thr2}$, $H_{thr1}$ was varied within the range [0.01, 0.1], and $H_{thr2}$ was within the range [0.2, 0.3]. Precision, recall, and the F1 score were calculated for different values of $H_{thr1}$ and $H_{thr2}$, as shown in Table 4 and Table 5. When $H_{thr1}$ and $H_{thr2}$ varied within these ranges, precision remained stable around 0.9, recall remained stable around 0.8, and the F1 score varied between 0.85 and 0.9.

The experimental results indicate that the algorithm’s extraction performance is not sensitive to the values of $H_{thr1}$ and $H_{thr2}$. Within the given ranges, the choice of $H_{thr1}$ and $H_{thr2}$ has a minimal impact on the algorithm, demonstrating good robustness. Therefore, the midpoints of the given intervals, 0.05 for $H_{thr1}$ and 0.25 for $H_{thr2}$, were selected as the values for these parameters.

#### 3.3.2. $\theta_{thr}$

$\theta_{thr}$ measures the degree to which the normal vectors of the point cloud points are perpendicular to the Z-axis direction vector. If $\theta_{thr}$ is set too low, some curbs that are not perfectly vertical to the Z-axis will be filtered out. Conversely, if $\theta_{thr}\;$ is set too high, a large number of noise points will pass through the filter.

To analyze the relationship between the extraction results and the values of $\theta_{thr}$, $\theta_{thr}$ was varied within the range [30°, 40°]. The resulting precision, recall, and F1 score values are shown in Table 6. When $\theta_{thr}$ varied within the range of [30°, 40°], precision, recall, and the F1 score remained stable within the ranges of [0.95, 1], [0.75, 0.85], and [0.8, 0.9], respectively. This indicates that the algorithm’s extraction performance is not sensitive to the variation of $\theta_{thr}$ within the given range, and the extraction results are weakly dependent on the choice of $\theta_{thr}$ threshold. Therefore, the midpoint of the given interval, 35°, was selected as the optimal value for $\theta_{thr}$.

#### 3.3.3. $D_{thr}$

$D_{thr}$ measures the lower limit of intra-cluster distance. If $D_{thr}$ is set too low, more clusters of obstacles will be retained. Conversely, if $D_{thr}$ is set too high, some shorter curbs will be filtered out, compromising the completeness of curb extraction.

To analyze the relationship between the extraction results and the values of $D_{thr}$, $D_{thr}$ was varied within the range [1.5, 2.5]. The resulting precision, recall, and F1 score values are shown in Table 7. When $D_{thr}$ varied within this range, precision, recall, and the F1 score remained stable within the ranges of [0.95, 1], [0.75, 0.85], and [0.85, 0.95], respectively. This indicates that the algorithm’s extraction performance is not sensitive to the variation of $D_{thr}$ within the given range, and the extraction results are weakly dependent on the choice of $D_{thr}$ threshold. Therefore, the midpoint of the given interval, 2 m, was selected as the optimal value for $D_{thr}$.

#### 3.3.4. $x_{0}$

$x_{0}$ is the lower limit of the MSAC distance threshold. $x_{0}$ will affect the weighting of each point to the fitted plane, leading to a discrepancy between the optimal fitted plane and the true curb plane.

To analyze the relationship between the extraction results and the values of $x_{0}$, $x_{0}$ was varied within the range [0.05, 0.15]. The resulting precision, recall, and F1 score values are shown in Table 8. When $x_{0}$ increased from 0.05 to 0.15, precision remained stable within the range [0.95, 1]. When $x_{0}$ was within the range of 0.05 to 0.07, recall was lower than 0.7, indicating a certain degree of impact on the extraction results. When $x_{0}$ was within the range of 0.08 to 0.15, recall remained within the range [0.75, 0.85], ensuring the completeness of curb extraction. The overall F1 score was stable around the range [0.8, 0.9], indicating the good overall extraction performance of the model. To ensure the accuracy and completeness of curb extraction, the midpoint of the interval [0.08, 0.15], which was 0.12, was selected as the optimal value for $x_{0}$.

## 4. Conclusions

This paper has presented a novel curb extraction method for unordered point clouds based on multi-feature filtering. The method begins with a coarse curb extraction using the maximum height difference of grids and normal vectors. To enhance precision, completeness, and robustness, it incorporates innovative steps such as clustering, a density-based alpha-shape algorithm, and a multi-frame fitted MSAC (M-Estimate Sample Consensus) filter. The multi-frame fitting approach addresses the limitations of traditional single-frame methods by fitting the curb contour every five frames, ensuring more accurate contour extraction while preserving local curb features.

We evaluated the method’s performance using data from both our self-developed dataset and the publicly available Toronto dataset, covering typical and complex road scenes. Quantitative analysis was conducted on five typical road scenarios: intersections, a straight road, curved roads, and ghosting. The experimental results demonstrate that the average curb segmentation F1 score is above 0.85, indicating high accuracy and robustness.

Additionally, a sensitivity analysis was performed for each parameter of the proposed method, providing desirable parameter ranges that did not significantly affect curb segmentation results. For example, maintaining the MSAC distance threshold between 0.08 m and 0.15 m ensures the completeness of curb extraction. The results show that the proposed method effectively overcomes interference in complex road environments, achieving accurate and complete curb extraction.

Future work will focus on addressing challenges related to segmenting and matching more than two road curbs simultaneously as the current curb matching rules have been designed for up to two curbs.

## Figures and Tables

**Figure 1 sensors-24-06544-f001:**
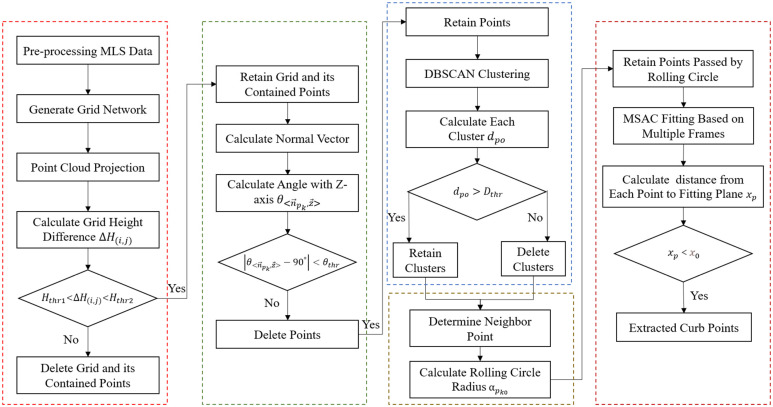
Algorithm framework.

**Figure 2 sensors-24-06544-f002:**
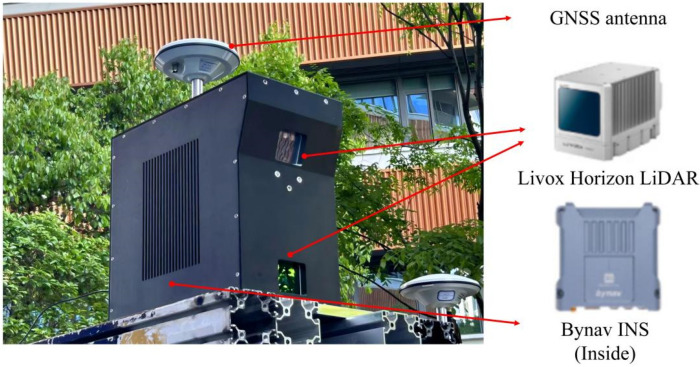
MLS system for field data acquisition.

**Figure 3 sensors-24-06544-f003:**
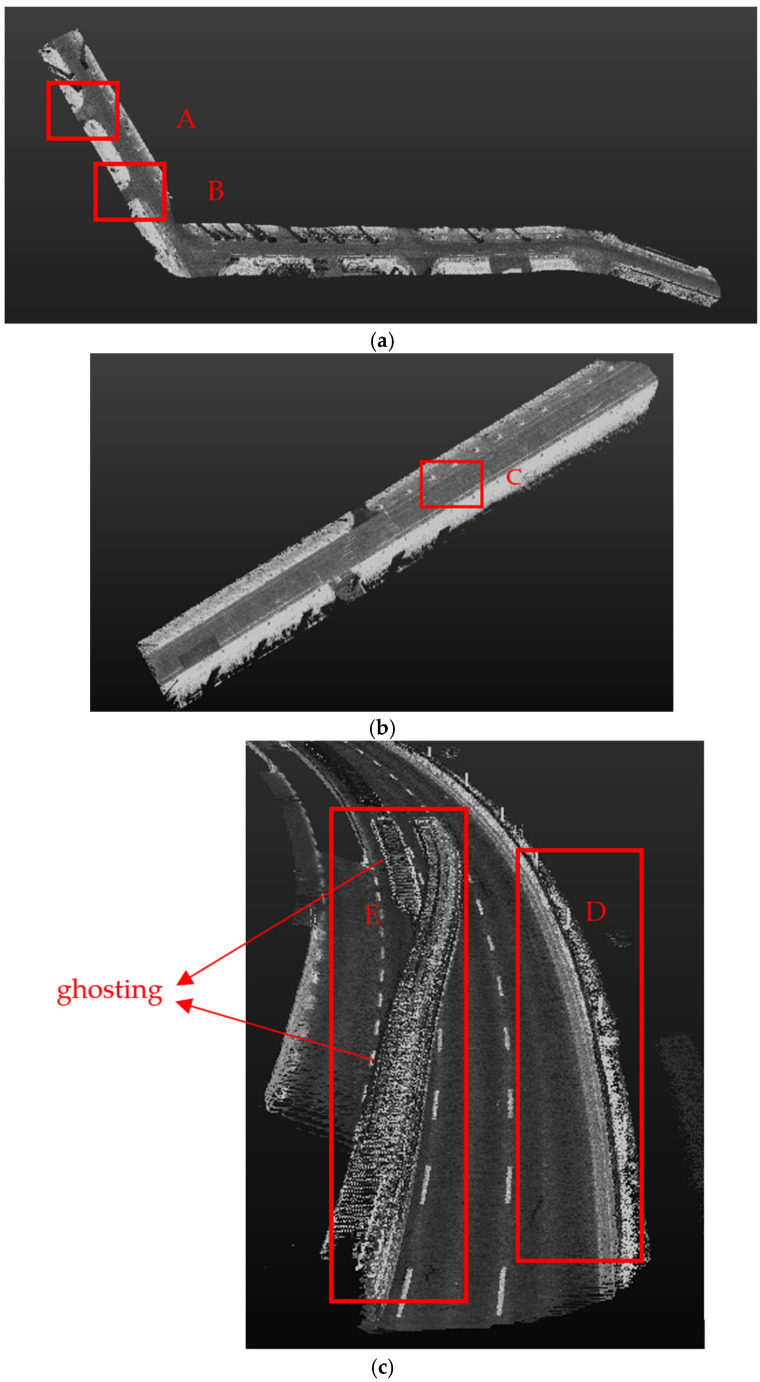
Various road scenarios. The road curb within the red box ABCDE is the scene for the comparison experiments.

**Figure 4 sensors-24-06544-f004:**
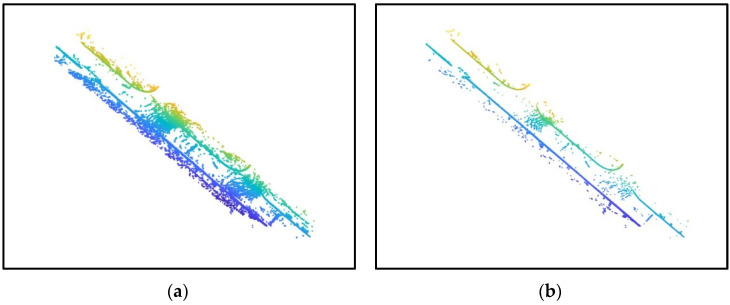
Extraction results for road scenarios A and B: (**a**) results after processing with grid height difference; (**b**) result after normal vector extraction; (**c**) result after using clustering and variable-radius alpha-shape algorithm; (**d**) result after multi-frame fitting MSAC algorithm. The different colors in the figure indicate varying heights of the curb points.

**Figure 5 sensors-24-06544-f005:**
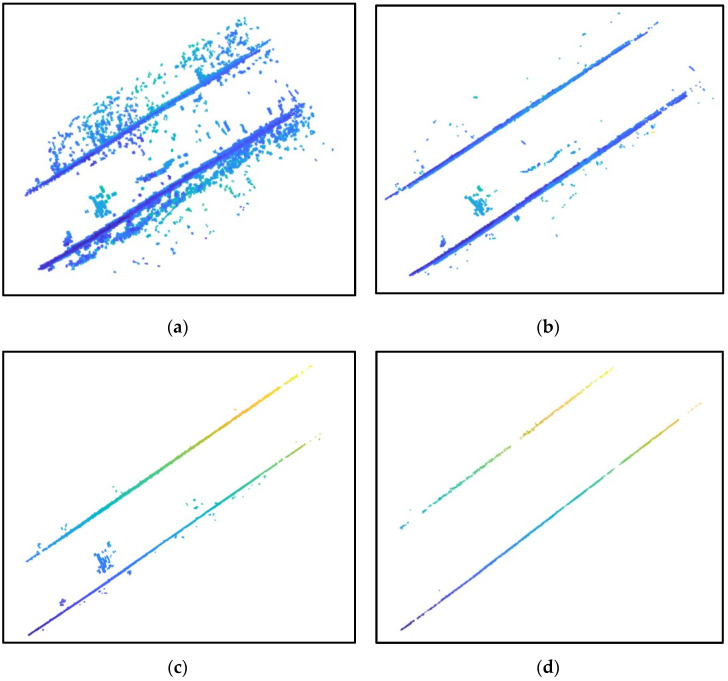
Extraction results for road scenario C: (**a**) result after processing with grid height difference; (**b**) result after normal vector extraction; (**c**) result after using clustering and variable-radius alpha-shape algorithm; (**d**) result after using multi-frame fitting MSAC algorithm.

**Figure 6 sensors-24-06544-f006:**
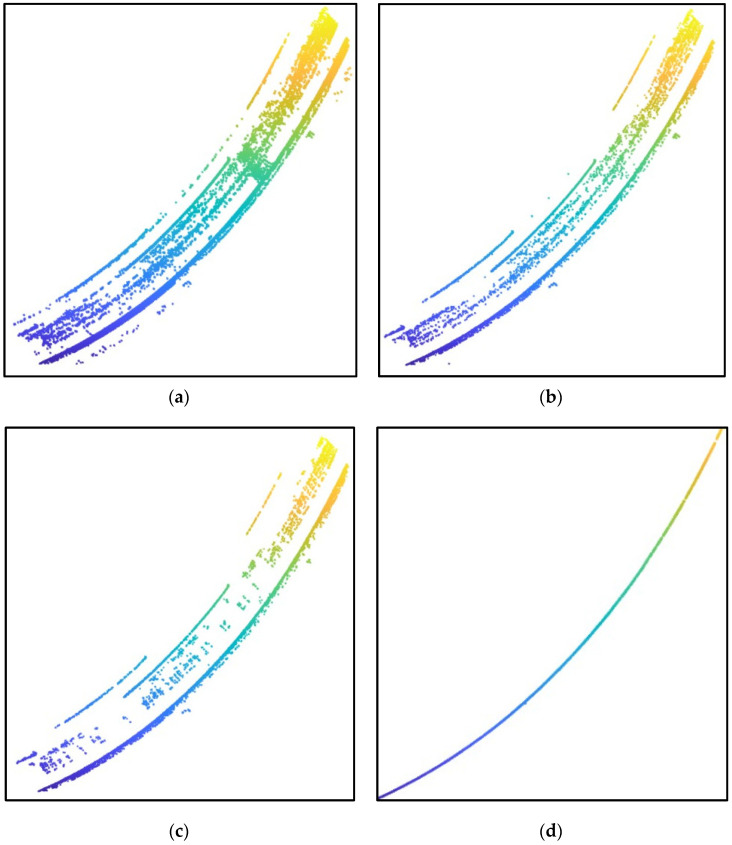
Extraction results for road scenarios D and E: (**a**) result after processing with grid height difference; (**b**) result after normal vector extraction; (**c**) result after using clustering and variable-radius alpha-shape algorithm; (**d**) result after using multi-frame fitting MSAC algorithm.

**Figure 7 sensors-24-06544-f007:**
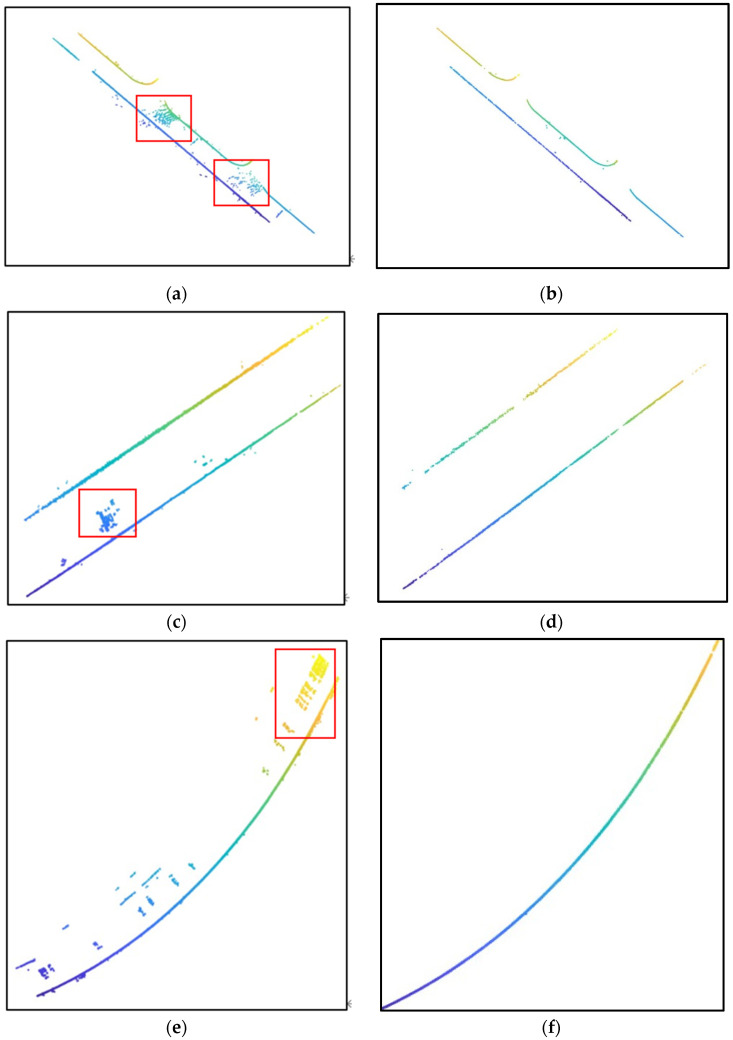
Comparison of road curb extraction using single-frame and multi-frame fitting: (**a**) road extraction result using single-frame fitting for road scenarios A and B; (**b**) road extraction result using multi-frame fitting for road scenarios A and B; (**c**) road extraction result using single-frame fitting for road scenarios C; (**d**) road extraction result using multi-frame fitting for road scenario C; (**e**) road extraction result using single-frame fitting for road scenarios D and E; (**f**) road extraction result using multi-frame fitting for road scenarios D and E.

**Figure 8 sensors-24-06544-f008:**
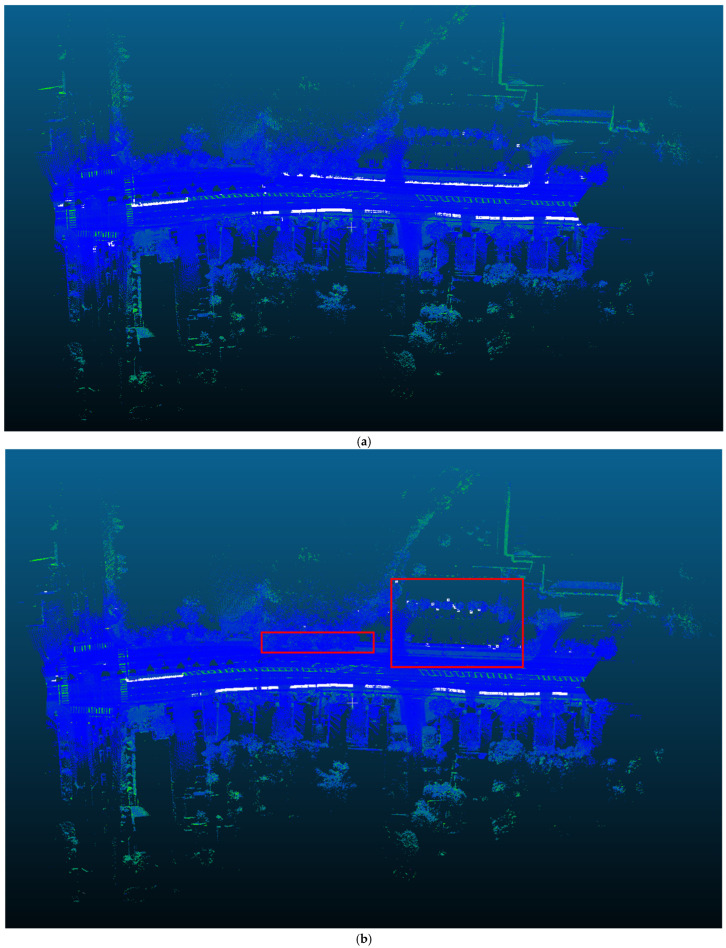
Comparison using the Toronto dataset: (**a**) our results using the Toronto dataset; (**b**) Mi’s results using the Toronto dataset.

**Table 1 sensors-24-06544-t001:** Parameter selection.

Parameter	Value	Description
$H_{thr1}$	0.05 m	Lower limit of grid height difference
$H_{thr2}$	0.25 m	Upper limit of grid height difference
$\theta_{thr}$	35°	Upper limit of the angle
$D_{thr}$	2 m	Lower limit of intra-cluster distance
$f_{0}$	5	Number of frames for multi-frame fitting
$x_{0}$	0.12 m	Lower limit of MSAC distance

**Table 2 sensors-24-06544-t002:** Quantitative results.

Scenario	Method	Precision	Recall	F1 Score
A, B	Mi’s [31]	0.8594	0.8337	0.8464
A, B	Ours	0.9542	0.7835	0.8605
C	Mi’s [31]	0.9125	0.8428	0.8762
C	Ours	0.9582	0.7985	0.8711
D, E	Mi’s [31]	0.8659	0.7365	0.7960
D, E	Ours	0.8972	0.7640	0.8252

**Table 3 sensors-24-06544-t003:** Quantitative results for the Toronto dataset.

Scenario	Method	Precision	Recall	F1 Score
Toronto	Mi’s. [31]	0.8272	0.7685	0.7968
Toronto	Ours	0.9285	0.8521	0.8889

**Table 4 sensors-24-06544-t004:** Sensitivity analysis of $H_{thr1}$.

$H_{thr1}$/m	Precision	Recall	F1 Score
0.01	0.8836	0.8162	0.8486
0.02	0.8873	0.7963	0.8393
0.03	0.9145	0.8272	0.8686
0.04	0.9561	0.7885	0.8643
0.05	0.9542	0.7835	0.8605
0.06	0.9604	0.8006	0.8732
0.07	0.9621	0.8382	0.8959
0.08	0.9621	0.774	0.8578
0.09	0.9593	0.7452	0.8388
0.1	0.9606	0.753	0.8442

**Table 5 sensors-24-06544-t005:** Sensitivity analysis of $H_{thr2}$.

H2/m	Precision	Recall	F1 Score
0.2	0.9531	0.7505	0.8398
0.21	0.9623	0.7619	0.8505
0.22	0.9533	0.8116	0.8767
0.23	0.9611	0.7811	0.8618
0.24	0.9545	0.8414	0.8944
0.25	0.9542	0.7835	0.8605
0.26	0.9505	0.879	0.9133
0.27	0.9562	0.7818	0.8602
0.28	0.9493	0.8641	0.9047
0.29	0.96	0.8013	0.8735
0.3	0.9565	0.8435	0.8965

**Table 6 sensors-24-06544-t006:** Sensitivity analysis of $\theta_{thr}$.

$\theta_{thr}$/°	Precision	Recall	F1 Score
30	0.9635	0.7211	0.8248
31	0.9616	0.72	0.8235
32	0.9568	0.7541	0.8434
33	0.96	0.741	0.8364
34	0.9613	0.7402	0.8364
35	0.961	0.7956	0.8705
36	0.9565	0.8733	0.913
37	0.9589	0.8453	0.8985
38	0.9594	0.8141	0.8808
39	0.9468	0.8463	0.8938
40	0.9574	0.8698	0.9115

**Table 7 sensors-24-06544-t007:** Sensitivity analysis of $D_{thr}$.

$D_{thr}$/m	Precision	Recall	F1 Score
1.5	0.9497	0.8779	0.9124
1.6	0.9607	0.7984	0.8721
1.7	0.963	0.775	0.8588
1.8	0.965	0.7818	0.8638
1.9	0.9605	0.8467	0.9
2	0.9542	0.7835	0.8605
2.1	0.9539	0.8435	0.8953
2.2	0.9597	0.7942	0.8691
2.3	0.9605	0.8275	0.8891
2.4	0.9601	0.7931	0.8686
2.5	0.9576	0.7622	0.8488

**Table 8 sensors-24-06544-t008:** Sensitivity analysis of $x_{0}$.

$x_{0}$/m	Precision	Recall	F1 Score
0.05	0.965	0.6561	0.7812
0.06	0.9657	0.6987	0.8108
0.07	0.9621	0.7204	0.8239
0.08	0.9617	0.7672	0.8535
0.09	0.9618	0.8048	0.8764
0.1	0.9542	0.7835	0.8605
0.11	0.9598	0.8396	0.8957
0.12	0.9506	0.7924	0.8643
0.13	0.9545	0.8705	0.9105
0.14	0.9536	0.8744	0.9123
0.15	0.9517	0.8943	0.9221

## Data Availability

Some or all data, models, or codes that support the findings of this study are available from the corresponding author upon reasonable request.

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
