# Peer review of "Multi-Feature-Filtering-Based Road Curb Extraction from Unordered Point Clouds"

_sensors, 2024, doi:10.3390/s24206544_

Round 1
Reviewer 1 Report
Comments and Suggestions for Authors
The paper proposes a method for extracting road curb points from unordered MLS point cloud data. The proposed method comprises several well-known algorithms such as DBSCAN, alpha-shape, and MSAC. The paper has the following issues:
1. The motivation is unsubstantial. For example, in Lines 118-119, the paper states that “current curb point cloud extraction methods are based on ordered point clouds,” and in Lines 131-132, it states that “most current research on curb extraction using point clouds is based on ordered point clouds.” However, no references about ordered point clouds are provided throughout the paper. In contrast, to my knowledge, many current curb point cloud extraction methods have already been proposed for unordered point clouds, such as [1], [12], [30].
2. No comparative experiments are provided in the paper. The proposed method should be compared to similar methods, such as [1], [12], [30].
3. The proposed method has only been tested on private MLS data. Some open MLS datasets should be included, such as KITTI 3D and Toronto 3D.
REFERENCES:
KITTI 3D: Geiger, Andreas, Philip Lenz, and Raquel Urtasun. "Are we ready for autonomous driving? the kitti vision benchmark suite." 2012 IEEE conference on computer vision and pattern recognition. IEEE, 2012.
Toronto 3D: Tan, Weikai, et al. "Toronto-3D: A large-scale mobile LiDAR dataset for semantic segmentation of urban roadways." Proceedings of the IEEE/CVF conference on computer vision and pattern recognition workshops. 2020.
Author Response
Reviewer#1 Comments
The paper proposes a method for extracting road curb points from unordered MLS point cloud data. The proposed method comprises several well-known algorithms such as DBSCAN, alpha-shape, and MSAC. The paper has the following issues:
- The motivation is unsubstantial. For example, in Lines 118-119, the paper states that “current curb point cloud extraction methods are based on ordered point clouds,” and in Lines 131-132, it states that “most current research on curb extraction using point clouds is based on ordered point clouds.” However, no references about ordered point clouds are provided throughout the paper. In contrast, to my knowledge, many current curb point cloud extraction methods have already been proposed for unordered point clouds, such as [1], [12], [30]..
Response:
Thank you for your inquiry. In line 130, we have added references to literatures [3, 15, 32, 33] on ordered point clouds. Although there have been methods for curb extraction using unordered point clouds, these methods do not perform optimally in certain specific scenarios, such as detecting ghost images and curved roads. Based on our self-developed unordered LiDAR point cloud dataset, the method proposed in this paper demonstrates robust performance in complex scenarios. Figure 1 and Figure 2 illustrate the point cloud data collected from an 0.8 km curved road segment with vehicle "ghosting" interference, where E represents the segment with ghosting interference. As indicated by D, this road segment has a curb on only one side, and the curb is located on the curved segment. Additionally, the comparative experiments (Table 3, Table 4, and Figure 8) with other methods and the performance analysis on the publicly available Toronto dataset, as introduced later, also support this conclusion.
[3] Sui, J. Zhu, M. Zhong, X. Wang, J. Kang, Extraction of road boundary from MLS data using laser scanner ground trajectory, Open Geosci. 2021, 13, 690–704.
[15] Yang, M., Liu, X., Jiang, K., Xu, J., Sheng, P., Yang, D. Automatic Extraction of Structural and Non-Structural Road Edges from Mobile Laser Scanning Data. Sensors, 2019, 19(22): 5030. https://doi.org/10.3390/s19225030.
[32] Guan, H., Li, J., Yu, Y., Chapman, M. and Wang, C., 2014. Automated road information extraction from mobile laser scanning data. IEEE Transactions on Intelligent Transportation Systems, 16(1), pp.194-205.
[33] Husain, A. and Vaishya, R.C., 2018. Road surface and its center line and boundary lines detection using terrestrial Lidar data. The Egyptian Journal of Remote Sensing and Space Science, 21(3), pp.363-374.
- No comparative experiments are provided in the paper. The proposed method should be compared to similar methods, such as [1], [12], [30].
Response:
Thank you for the expert's questions. We have added the comparison methods and the quantitative analysis on the Toronto dataset. In line 474, comparisons with Mi’s method [30], have been added. The results indicate that the method proposed in this paper ensures good performance in some complex scenarios. Mi’s method uses Supervoxel generation technique for road boundary extraction from unstructured mobile laser scanning (MLS) point clouds, followed by Kalman filter tracking and a refine operator to maintain road boundaries completeness and accuracy. This method, used for comparison, leverages the unordered characteristics of the dataset, ensuring fairness and validity in the comparison. Additionally, it takes into account curved roads and other complex scenarios. Being published in 2022, it also ensures the novelty of the algorithm comparison.
[30] Mi, X., Yang, B., Dong, Z., Chen, C., Gu, J. Automated 3D Road Boundary Extraction and Vectorization Using MLS Point Clouds. IEEE Transactions on Intelligent Transportation Systems, 2022, 23(6): 5287–5297.
As shown in Table 1, the performance of the method proposed in this paper is compared with that of Mi’s method across scenarios A to E. In relatively simple scenarios, such as the straight-line scenario C, the F1 score of the proposed method is slightly lower than Mi's method [30], with only a small difference in results. However, in more complex scenarios, such as A and B (intersections) and D and E (curved roads and ghosting), the F1 score of the proposed method is higher, with better road edge extraction performance. This demonstrates the strong robustness of the proposed method in handling complex scenes.
Table 1 (referred to as Table 3 in the paper). Quantitative Results
|
Scenario |
Method |
Precision |
Recall |
F1-score |
|
A、B |
Mi’s. [30] |
0.8594 |
0.8337 |
0.8464 |
|
A、B |
Ours |
0.9542 |
0.7835 |
0.8605 |
|
C |
Mi’s. [30] |
0.9125 |
0.8428 |
0.8762 |
|
C |
Ours |
0.9582 |
0.7985 |
0.8711 |
|
D、E |
Mi’s. [30] |
0.8659 |
0.7365 |
0.7960 |
|
D、E |
Ours |
0.8972 |
0.7640 |
0.8252 |
To further validate the effectiveness of the proposed method, additional experiments were conducted on the publicly available Toronto dataset. A 3 km road section was selected for testing, comparing both the proposed method and Mi's method. The results, shown in Table 2, indicate that our method outperforms Mi’s method in both precision and recall, demonstrating its ability to accurately and completely extract curbs.
Table 2 (referred to as Table 4 in the paper). Quantitative Results on the Toronto dataset
|
Scenario |
Method |
Precision |
Recall |
F1-score |
|
Toronto |
Mi’s. [30] |
0.8872 |
0.8375 |
0.8616 |
|
Toronto |
Ours |
0.9285 |
0.8521 |
0.8889 |
Figure 3(referred to as Figure 8 in the paper). Comparison on the Toronto dataset: (a) ours result on the Toronto; (b) Mi’s result on the Toronto
- The proposed method has only been tested on private MLS data. Some open MLS datasets should be included, such as KITTI 3D and Toronto 3D.
Response:
Thank you for your question. The datasets we selected represent common road curb scenarios, including straight-line, curved, corner, and intersection scenarios, to demonstrate the effectiveness and robustness of the algorithm. Toronto is a relatively new dataset, and unlike KITTI, which has distinct scanning line characteristics, Toronto more closely resembles the point cloud data collected in this paper. For this reason, we chose Toronto as the comparative dataset and added a set of comparative experiments based on it.

Reviewer 2 Report
Comments and Suggestions for Authors
Road curb extraction from unordered point clouds has been studied for many years. In this paper, firstly, a brief introduction to 3D point cloud networking and 3D feature-based road curb extraction is given, and secondly, how to inhibit the extraction of 3D disordered point clouds in complex scenes as a scientific problem, using network height difference and normal vector features as the first step of coarse extraction, and clustering, alpha-shape algorithm based on density, and MSAC algorithm based on multi-frame fitting as the second step for fine extraction of edge points.
The current form of the paper should be carefully revised. The title and contribution of the paper need to be further condensed, and there are some specific issues that need to be fixed.
· Abstract: There is no need to clarify the “Background”, “Methods”,” Results and Conclusions”. Line 20-22: the alpha-shape algorithm and the MSAC (M-Estimate Sample Consensus) algorithm are not features. The abstract needs to be rewritten.
· Line 16-17: There have also been many methods in the past for unordered point clouds. The author's question is too broad.
· Figure 2: There is no complete onboard LiDAR system available.
· Experimental Data: Experimental data should be provided for experimental design purposes. It is hard to see the representativeness of the three various road scenarios.
· There are already many traditional methods, for which comparative experiments should have been added.
· Related Work and References: Relevant literature should be thoroughly investigated.
Author Response
Reviewer #2 Comments:
- The current form of the paper should be carefully revised. The title and contribution of the paper need to be further condensed, and there are some specific issues that need to be fixed.
Response:
Thank you very much for your suggestion. We have revised the title to "Multi-Feature Filtering-Based Road Curb Extraction from Unordered Point Clouds" to better reflect the core methodology and logic of the paper. This paper presents a novel curb extraction method for unordered point clouds that focuses on multi-feature filtering. The method integrates several techniques, including grid height difference, normal vectors, clustering, an alpha-shape algorithm based on point cloud density, and the MSAC (M-Estimate Sample Consensus) algorithm for multi-frame fitting. Together, these techniques form a robust filter capable of accurately identifying curbs in various complex scenarios.
Our team believes this modification aligns more closely with the paper's overall narrative, emphasizing the process of filtering and selecting relevant features to accurately and comprehensively extract curb points in different road environments. We also believe this change will enhance the paper's readability and clarity for the audience, while highlighting the effectiveness and robustness of the proposed method.
Based on the newly added comparative experiments with [30] and validation using the publicly available Toronto dataset, we have also revised the abstract and contributions. The specific changes are as follows:
Abstract:
Road curb extraction is a critical component of road environment perception, essential for calculating road geometry parameters and ensuring the safe navigation of autonomous vehicles. Existing research primarily focuses on extracting curbs from ordered point clouds, which are constrained by their structure of the point cloud organization, making them difficult to apply to unordered point cloud data and susceptible to interference from obstacles. To overcome these limitations, a multi-feature filtering-based method for curb extraction from unordered point clouds is proposed. This method integrates several techniques, including grid height difference, normal vectors, clustering, an alpha-shape algorithm based on point cloud density, and the MSAC (M-Estimate Sample Consensus) algorithm for multi-frame fitting. The multi-frame fitting approach addresses limitations of traditional single-frame methods by fitting the curb contour every 5 frames, ensuring more accurate contour extraction while preserving local curb features. Based on our self-developed dataset and the Toronto dataset, these methods are integrated to create a robust filter capable of accurately identifying curbs in various complex scenarios. Optimal threshold values were determined through sensitivity analysis and applied to enhance curb extraction performance under diverse conditions. Experimental results demonstrate that the proposed method accurately and comprehensively extracts curb points in different road environments, proving its effectiveness and robustness. Specifically, the average curb segmentation Precision, Recall, and F1-score across scenarios A, B (intersections), C (straight road), and D, E (curved roads and ghosting) are 0.9365, 0.782, and 0.8523, respectively.
Contributions:
- Proposed a Multi-Feature Filtering framework for curb extraction from unordered point clouds. This framework integrates several techniques, including grid height difference, normal vectors, clustering, a density-based alpha-shape algorithm, and a multi-frame fitting approach using the MSAC algorithm to accurately identify curbs in complex road scenarios.
- Introduced a multi-frame MSAC fitting approach, which improves on traditional single-frame methods by fitting the curb contour every 5 frames. This approach captures the full curb contour more accurately while preserving local features. Based on our self-developed dataset and the Toronto dataset, our method outperforms existing approaches, such as Mi's method, in both precision and recall, demonstrating its robustness across various complex road scenarios.
- Performed parameter sensitivity analysis across different road scenarios to determine the range of parameter values that do not significantly influence the curb segmentation results, further validating the robustness of the method.
Conclusion:
This paper presents a novel curb extraction method for unordered point clouds based on multi-feature filtering. The method begins with a coarse curb extraction using the maximum height difference of grids and normal vectors. To enhance precision, completeness, and robustness, it incorporates innovative steps such as clustering, a density-based alpha-shape algorithm, and a multi-frame fitted MSAC (M-Estimate Sample Consensus) filter. The multi-frame fitting approach addresses limitations of traditional single-frame methods by fitting the curb contour every 5 frames, ensuring more accurate contour extraction while preserving local curb features.
We evaluated the method's performance using data from both our self-developed dataset and the publicly available Toronto dataset, covering typical and complex road scenes. Quantitative analysis was conducted on five typical road scenarios: intersections, a straight road, curved roads, and ghosting. The experimental results demonstrate that the average curb segmentation F1-score is above 0.85, indicating high accuracy and robustness.
Additionally, a sensitivity analysis was performed for each parameter of the proposed method, providing desirable parameter ranges that do not significantly affect curb segmentation results. For example, maintaining the MSAC distance threshold between 0.08 m and 0.15 m ensures the completeness of curb extraction. The results show that the proposed method effectively overcomes interference in complex road environments, achieving accurate and complete curb extraction.
Future work will focus on addressing challenges related to segmenting and matching more than two road curbs simultaneously, as the current curb matching rules are designed for up to two curbs.
- 2. Abstract: There is no need to clarify the “Background”,“Methods”, “Results and Conclusions”. Line 20-22: the alpha-shape algorithm and the MSAC (M-Estimate Sample Consensus) algorithm are not features. The abstract needs to be rewritten.
Response:
Thank you very much for your suggestions. We have removed the “Background”, “Methods”,” Results and Conclusions” from the abstract. We have changed “features” to “methods” in lines 20-22.
- 3.Figure 2: There is no complete onboard LiDAR system available.
Response:
Thank you for your valuable feedback. We have revised Figure 2 and the description of the data collection. The updated details are as follows:
Figure 2 illustrates the self-constructed MLS system for field data acquisition [1, 4]. The Livox Horizon 3D LiDAR (hereafter referred to as Horizon) is used to collect road point cloud data. The LiDAR has a wavelength of 905 nm, a horizontal field of view (FOV) of 81.7°, and a vertical FOV of 25.1°. Horizon utilizes Livox's independently developed high-speed non-repetitive scanning technology and a custom-designed multi-line packaged laser, enabling rapid capture of scene details. Horizon can be configured in single/double echo modes, with a point cloud data rate reaching up to 480,000 points per second in double echo mode. Horizon's adaptability to the environment is strong; even under intense sunlight interference of 100 klx, the noise rate remains below 0.01%. The INS sensor provides essential GPS positioning and attitude data. The MLS system is mounted on a moving vehicle to scan road environment information and outputs the obtained road point cloud.
The specific data collection setup and scene requirements can be referenced from our team's previously published research [1, 4].
Figure 2. MLS system for field data acquisition
- 4. Experimental Data: Experimental data should be provided for experimental design purposes. It is hard to see the representativeness of the three various road scenarios.
Response:
We have revised the experiments are designed to demonstrate that the algorithm proposed in this paper can still maintain high recognition accuracy in complex road environments. The scenarios proposed in this paper include straight-line scenarios, curved scenarios, corner scenarios, and intersection scenarios, which basically cover the common road curb scenarios. Additionally, in line 477, we also conducted comparative experiments of our method on the public dataset Toronto, and the results show that our proposed method can accurately and completely extract road curbs.
- There are already many traditional methods, for which comparative experiments should have been added.
Response:
Thank you for the expert's questions. We have added the comparison methods and the quantitative analysis on the Toronto dataset. In line 474, comparisons with Mi’s method [30], have been added. The results indicate that the method proposed in this paper ensures good performance in some complex scenarios. Mi’s method uses Supervoxel generation technique for road boundary extraction from unstructured mobile laser scanning (MLS) point clouds, followed by Kalman filter tracking and a refine operator to maintain road boundaries completeness and accuracy. This method, used for comparison, leverages the unordered characteristics of the dataset, ensuring fairness and validity in the comparison. Additionally, it takes into account curved roads and other complex scenarios. Being published in 2022, it also ensures the novelty of the algorithm comparison.
[30] Mi, X., Yang, B., Dong, Z., Chen, C., Gu, J. Automated 3D Road Boundary Extraction and Vectorization Using MLS Point Clouds. IEEE Transactions on Intelligent Transportation Systems, 2022, 23(6): 5287–5297.
As shown in Table 1, the performance of the method proposed in this paper is compared with that of Mi’s method across scenarios A to E. In relatively simple scenarios, such as the straight-line scenario C, the F1 score of the proposed method is slightly lower than Mi's method [30], with only a small difference in results. However, in more complex scenarios, such as A and B (intersections) and D and E (curved roads and ghosting), the F1 score of the proposed method is higher, with better road edge extraction performance. This demonstrates the strong robustness of the proposed method in handling complex scenes.
Table 1 (referred to as Table 3 in the paper). Quantitative Results
|
Scenario |
Method |
Precision |
Recall |
F1-score |
|
A、B |
Mi’s. [30] |
0.8594 |
0.8337 |
0.8464 |
|
A、B |
Ours |
0.9542 |
0.7835 |
0.8605 |
|
C |
Mi’s. [30] |
0.9125 |
0.8428 |
0.8762 |
|
C |
Ours |
0.9582 |
0.7985 |
0.8711 |
|
D、E |
Mi’s. [30] |
0.8659 |
0.7365 |
0.7960 |
|
D、E |
Ours |
0.8972 |
0.7640 |
0.8252 |
To further validate the effectiveness of the proposed method, additional experiments were conducted on the publicly available Toronto dataset. A 3 km road section was selected for testing, comparing both the proposed method and Mi's method. The results, shown in Table 2, indicate that our method outperforms Mi’s method in both precision and recall, demonstrating its ability to accurately and completely extract curbs.
Table 2 (referred to as Table 4 in the paper). Quantitative Results on the Toronto dataset
|
Scenario |
Method |
Precision |
Recall |
F1-score |
|
Toronto |
Mi’s. [30] |
0.8872 |
0.8375 |
0.8616 |
|
Toronto |
Ours |
0.9285 |
0.8521 |
0.8889 |
Figure 3(referred to as Figure 8 in the paper). Comparison on the Toronto dataset: (a) ours result on the Toronto; (b) Mi’s result on the Toronto
- 6. Related Work and References: Relevant literature should be thoroughly investigated.
Response:
Thank you very much for your inquiry. We have further reviewed the relevant literature and made corresponding changes to the article based on the results of our reading.
3D point clouds are categorized into ordered and unordered types. Current curb point cloud extraction methods are based on ordered point clouds. Ordered point clouds have points organized in a specific sequence, with each point's position information associated with its index in the point cloud, having a clear topological structure. They are usually used to represent structured data obtained through scanning or modeling [3, 15, 32, 33].
- Guan, H., Li, J., Yu, Y., Chapman, M. and Wang, C., 2014. Automated road information extraction from mobile laser scanning data. IEEE Transactions on Intelligent Transportation Systems, 16(1), pp.194-205.
- Husain, A. and Vaishya, R.C., 2018. Road surface and its center line and boundary lines detection using terrestrial Lidar data. The Egyptian Journal of Remote Sensing and Space Science, 21(3), pp.363-374.

Round 2
Reviewer 1 Report
Comments and Suggestions for Authors
The paper has been revised accordingly.
Reviewer 2 Report
Comments and Suggestions for Authors
I accept the current overall form, but I suggest conducting a grammar check on it.
Comments on the Quality of English LanguageI suggest conducting a grammar check on it.